# Zingerone in the Flower of *Passiflora maliformis* Attracts an Australian Fruit Fly, *Bactrocera jarvisi* (Tryon)

**DOI:** 10.3390/molecules25122877

**Published:** 2020-06-22

**Authors:** Soo Jean Park, Stefano G. De Faveri, Jodie Cheesman, Benjamin L. Hanssen, Donald N. S. Cameron, Ian M. Jamie, Phillip W. Taylor

**Affiliations:** 1Applied BioSciences, Faculty of Science and Engineering, Macquarie University, Sydney, NSW 2109, Australia; donald.cameron@mq.edu.au (D.N.S.C.); phil.taylor@mq.edu.au (P.W.T.); 2Horticulture and Forestry Science, Queensland Department of Agriculture and Fisheries, Mareeba, QLD 4880, Australia; stefano.defaveri@daf.qld.gov.au (S.G.D.F.); jodie.cheesman@daf.qld.gov.au (J.C.); 3Department of Molecular Sciences, Faculty of Science and Engineering, Macquarie University, Sydney, NSW 2109, Australia; benjamin.hanssen@hdr.mq.edu.au (B.L.H.); ian.jamie@mq.edu.au (I.M.J.)

**Keywords:** passion fruit flower, Jarvis’s fruit fly, phenylpropanoids, raspberry ketone, cuelure, GC-MS

## Abstract

*Passiflora maliformis* is an introduced plant in Australia but its flowers are known to attract the native Jarvis’s fruit fly, *Bactrocera jarvisi* (Tryon). The present study identifies and quantifies likely attractant(s) of male *B. jarvisi* in *P. maliformis* flowers. The chemical compositions of the inner and outer coronal filaments, anther, stigma, ovary, sepal, and petal of *P. maliformis* were separately extracted with ethanol and analyzed using gas chromatography-mass spectrometry (GC-MS). Polyisoprenoid lipid precursors, fatty acids and their derivatives, and phenylpropanoids were detected in *P. maliformis* flowers. Phenylpropanoids included raspberry ketone, cuelure, zingerone, and zingerol, although compositions varied markedly amongst the flower parts. *P. maliformis* flowers were open for less than one day, and the amounts of some of the compounds decreased throughout the day. The attraction of male *B. jarvisi* to *P. maliformis* flowers is most readily explained by the presence of zingerone in these flowers.

## 1. Introduction

Mature males of many dacine fruit flies (Tephritidae) are attracted to the flowers of certain plants and this attraction is often related to the presence of either methyl eugenol (4-allyl-1,2-dimethoxybenzene) or raspberry ketone (4-(4-hydroxyphenyl)-2-butanone) in these flowers [1,2,3,4,5,6]. For example, the flowers of most *Bulbophyllum* orchids do not produce nectar, but instead produce volatile compounds to attract pollinators [7], including *Bactrocera* fruit flies [7,8,9]. A broad range of *Bactrocera* species is attracted to methyl eugenol and cuelure (4-(4-acetoxyphenyl)-2-butanone), a more volatile analog of raspberry ketone [1,10,11]. These are the standard lures used in surveillance and monitoring traps and for male annihilation techniques. Zingerone (4-(4-hydroxy-3-methoxyphenyl)-2-butanone) contains common functional groups from both methyl eugenol and cuelure, and the zingerone-containing flowers of *Bulbophyllum patens* attract a wide range of methyl eugenol and cuelure responsive flies [12]. Male flies that feed on methyl eugenol, raspberry ketone, or cuelure are known to gain substantial increases in sexual performance [13,14,15,16,17].

*Bactrocera jarvisi* (Tryon) is a moderate pest fruit fly in Northern Australia, having a host range of 83 species, of which 15 are cultivated [18]. The distribution of *B. jarvisi* is largely consistent with its native host, *Planchonia careya* (Cocky apple) [19], extending down the east coast of Australia, to as far south as Sydney, and across northern Australia into the Northern Territory and to Broome in Western Australia [18,20]. While they exhibit little response to cuelure, *B. jarvisi* males are strongly attracted to zingerone [21,22]. *Bactrocera jarvisi* males are attracted to the flowers of an Australian native orchid, *Bu. baileyi*, which, like *Bu. patens*, contains zingerone [2,7,12]. The attraction of *B. jarvisi* to flowers of the native tar tree, *Semecarpus australiensis*, and two exotic (American) passion fruits, *Passiflora maliformis* and *P. ligularis*, has been reported [22,23], but compound(s) responsible for this attraction have not been identified.

*Passiflora maliformis* is native to the Caribbean, Central America, and northern South America [24]. Although *P. maliformis* is not cultivated commercially, it does produce edible fruits [25,26,27]. The seeds of *P. maliformis* contain essential fatty acids [25], and the fruit juice has antioxidant activity [26]. Chemical profiles of whole fruit extracts of *P. maliformis* have been studied, reporting six major and 32 minor components [28]. Headspace profiles of *P. maliformis* flowers are known to contain indole as the major compound, bisabolene, geranyl acetone, 6-methyl-5-hepten-2-one, and (*E*)-β-ocimene [22]. However, the reported headspace profiles do not include phenylpropanoid compound(s) that are known to attract *B. jarvisi* [22].

The flowers of *P. maliformis* open for only one day before senescence, and the authors have observed that more *B. jarvisi* males are attracted to the coronal filaments of the flower in the morning than later in the day. Male *Bactrocera* is commonly more responsive to lures in the early morning than other times of the day [29,30,31], and so this observation may simply reflect diurnal patterns of *B. jarvisi* lure response. However, it may also be that *P. maliformis* flowers release more attractant(s) in the morning.

To ascertain the chemical basis for attraction of *B. jarvisi* to flowers of *P. maliformis*, a species with which they do not share an evolutionary history, the present study (1) identifies compound(s) in *P. maliformis* flowers that might attract male *B. jarvisi*, and (2) quantifies prospective attractant(s) in the flowers through the day.

## 2. Results

### 2.1. Chemical Profiles of P. maliformis Flowers

The identified compounds and their location in *P. maliformis* flowers are shown in Table 1. The flowers contained terpenoids, C_16_ and C_18_ fatty acids and their derivatives, phenylpropanoids, farnesol, farnesyl acetate, and squalene. Fatty acids and their derivatives and farnesol derivatives were most abundant in the coronal filament extracts. Fatty acids and their derivatives are most abundant in the reproductive organs, while terpenoids are most abundant in the petal and sepal. Phenylpropanoids included raspberry ketone, cuelure, zingerone, and zingerol. These are present but minor in the coronal filaments. Traces of raspberry ketone, cuelure, and zingerone were detected in the petal and sepal. The compounds, 3-ethyl-2,3-dihydro-4*H*-pyran-4-one and 1-(4-methyl-1,3-dioxolan-2-yl)pentan-3-one were tentatively identified (See Appendix A in Appendix A).

### 2.2. Diurnal Changes in Raspberry Ketone, Cuelure, Zingerone and Zingerol in P. maliformis Coronal Filaments

Four known fruit fly attractants, raspberry ketone, cuelure, zingerone, and zingerol were detected. However, the compounds were not detected in all flower parts. The sepal and petal showed traces of raspberry ketone, cuelure, and zingerone, but as the amount was below the limit of detection of the generated standard curves, further quantification in these flower parts was not possible. Figure 1 illustrates changes in the amounts of the four compounds from 6 am to 5 pm AEST. Statistical analysis showed that in the inner coronal filament, the amounts of raspberry ketone (*F*_11,46_ = 1.0, *p* > 0.05), cuelure (*F*_11,46_ = 1.1, *p* > 0.05), and zingerol (*F*_11,46_ = 1.6, *p* > 0.05) did not change with flower collection time, but the amount of zingerone changed through the day (*F*_11,46_ = 6.4, *p* < 0.0001). In the outer coronal filaments, the amount of cuelure (*F*_11,46_ = 2.7, *p* < 0.01) and zingerone (*F*_11,46_ = 5.2, *p* < 0.0001) changed through the day, but the amount of raspberry ketone (*F*_11,46_ = 1.1, *p* > 0.05) and zingerol (*F*_11,46_ = 1.5, *p* > 0.05) did not change. Further analysis of non-linear least-squares fitting showed that the amounts of zingerone and cuelure decayed exponentially through the day (Table 2 and see residual plots in Appendix A in Appendix A). Although the data of the raspberry ketone in the inner coronal filament converged when fitted to the function, the fitting results showed that a decay pattern was not strong (Table 2 and Appendix A in Appendix A). The data of zingerol in both filaments, raspberry ketone in the outer coronal filament, and cuelure in the inner coronal filament did not converge to fit the function.

## 3. Discussion

The biosynthetic intermediates for polyisoprenoid lipids, such as farnesol, farnesyl acetate, and squalene, found in the flowers of *P. maliformis* are not unexpected as these are essential components involved in the mevalonate pathway in plants [47]. This pathway is also associated with the synthesis of terpenoids. Terpenoids, such as borneol, β-pinene, and terpin found in the current study, are structurally diverse and are the most common plant secondary metabolites [48]. Borneol and both α- and β-pinene are commonly found as volatile plant compounds [49,50,51]. Terpin is found in a bamboo species [52]. Release of volatile terpenoids, such as those reported in the present study, is implicated in the protection of plants against abiotic stresses, such as drought, high temperatures, and oxidative stresses [53,54], as well as biotic interactions, such as herbivores or pathogenic attacks [55]. The present study detected indole as a minor compound, while Fay reported it as the major compound along with several other compounds [22]. This inconsistency might have been due to the different sampling methods; the present study utilized solvent extraction of flower parts, while Fay entrapped headspace from cut vines with flowers. Given that biosynthesis and accumulation of volatiles are highly compartmentalized and restricted to specific tissues [56,57,58], and emission profiles can vary with flower conditions during the sampling [59,60], the inconsistency is not unprecedented. The occurrence of saturated and unsaturated fatty acids and their derivatives, including a suite of C_18_ fatty acid derivatives, is ubiquitous in plants [61]. These acids are building blocks for very long-chain fatty acids that are precursors for the synthesis of sphingolipids, important molecules involved in signal transduction and regulation of cell death [62].

The detected phenylpropanoids, including raspberry ketone, cuelure, zingerone, and zingerol in the coronal filaments confirms the previous speculation that the flower of *P. maliformis* may contain zingerone that attracts *B. jarvisi* males [21,22]. Some volatile phenylpropanoids are emitted as floral attractants to pollinators or as defense compounds against microorganisms, insects, and mammalian predators [63,64,65]. Likewise, the phenylpropanoids found in this exotic plant are floral attractants to *Bactrocera* species in Australia, although their functions in the plant’s native environment are not known.

The production and emission of volatile compounds in plants increase during the early stages of organ development and then either remain relatively constant or decrease over the organs’ lifespan [64]. This trend was observed in the present study in that the amounts of zingerone in both coronal filaments and cuelure in the outer coronal filament decayed exponentially, and that of the other compounds remained unchanged or did not show a strong decay pattern. The release of floral volatiles is also controlled by circadian clock [66,67], and environmental factors, such as light, temperature, and humidity [68]. The overall amounts of zingerone detected in the corona were higher in the inner filaments than in the outer filaments (Figure 1). Cuelure was known only as a synthetic attractant [1] until recent studies reported the occurrence of cuelure in nature [69,70,71]. The present study also demonstrates cuelure as a naturally occurring compound. If the flowers released sufficient cuelure and raspberry ketone, then cuelure-responding species should be attracted. The amount of cuelure and raspberry ketone in the flowers was barely detectable, which likely explains why the flowers have not been reported to attract cuelure-responding species. The attraction of *B. jarvisi* to zingerone is as strong as that of *B. dorsalis* to methyl eugenol [21]. The efficacy of cuelure to cuelure-responding species is not as strong as that of methyl eugenol to methyl eugenol responding species [72,73]. In combination with the very low levels of cuelure and raspberry ketone in the flowers, the weaker efficacy of cuelure and raspberry ketone may also explain why species that are attracted to these compounds have not been reported on *P. maliformis* flowers.

The release of floral volatiles in plants often coincides with the foraging activities of potential pollinators and is usually regulated by light [64]. Although there is variation in insect pollination between *Passiflora* species [74], generally, the pollinators for *Passiflora* include bees, hummingbirds, and bats [75,76,77]. The corona of *Passiflora* is known to act as a visual and olfactory stimulus to its pollinators [78]. Substances in the corona of *Passiflora*, including phenylpropanoids, may play a role in attracting pollinators. The diurnal patterns of zingerone and cuelure contents of *P. maliformis* flowers coincide with the typical diurnal pattern of responsiveness of *Bactrocera* males, which are usually more responsive to phenylpropanoid attractants in the early morning [29,30,31]. Further study is needed to confirm whether peak responsiveness of *B. jarvisi* to phenylpropanoids corresponds with the timing of the greatest amounts of phenylpropanoids in *P. maliformis*. However, such correspondence cannot be explained in evolutionary terms, as the natural range of *P. maliformis* does not coincide with that of any known *Bactrocera* species. The functional significance of zingerone and other phenylpropanoids in *P. maliformis* flowers and its significance to pollinators in the natural range of this plant is currently unclear.

In summary, the present study finds a cause of interaction between an exotic plant, *P. maliformis* and an Australian native fruit fly, *B. jarvisi* to be most likely by plant-produced zingerone, a volatile compound that is highly attractive to *B. jarvisi* and is also released by flowers of some native plants to attract insect pollinators.

## 4. Materials and Methods

### 4.1. Chemicals

For positive identification of the compounds in *P. maliformis* flowers, β-pinene, borneol, indole, terpin, raspberry ketone (4-(4-hydroxyphenyl)-2-butanone), cuelure (4-(4-acetoxyphenyl)-2-butanone), zingerone (4-(4-hydroxy-3-methoxyphenyl)-2-butanone), farnesol, palmitic acid, ethyl palmitate, phytol, linolenic acid, ethyl linoleate, ethyl linolenate, stearic acid, ethyl stearate, and squalene were purchased from Sigma-Aldrich (Merck KGaA, Darmstadt, Germany). Nonanoic acid was purchased from Acros Organics (Thermo Fisher Scientific, Geel, Belgium). Propyl laurate was purchased from AbovChem (AbovChem LLC, San Diego, CA, USA). Farnesyl acetate (a mixture of isomers) was purchased from TCI (Tokyo Chemical Industry Co., Ltd. Tokyo, Japan). Linoleic acid was purchased from Alfa-Aesar (Thermo Fisher Scientific, Lancashire, UK). The chemicals were analytical or reagent grade with 98% or higher purity and used without further purifications. Zingerol (4-(4-hydroxy-3-methoxyphenyl)-2-butanol) was synthesized by the reduction of zingerone with sodium borohydride in methanol (See Appendix A for synthetic details).

### 4.2. Flower Extraction

Flowers of *P. maliformis* (Voucher number: CNS 147961.1, Australian Tropical Herbarium) were collected between January 2017 and October 2018 at the Mareeba Research Facility of the Queensland Department of Agriculture and Fisheries (17.007007° S, 145.429446° E). The flowers were collected hourly from 6 am to 5 pm AEST with at least three replicates at each hour. Following collection, the inner and outer coronal filaments, anther, stigma, ovary, sepal, and petal were separated (see Appendix A in Appendix A for the organization of the flower organs), cut into fine pieces and stored in absolute ethanol (1 mL) in 2 or 4 mL glass vials at 4 °C until further extraction. Scissors and forceps were decontaminated with 100% ethanol in between dissecting and cutting the separated flower parts. The ethanol extracts were partitioned between aqueous and organic layers by adding water (1 mL) and ethyl acetate in hexane (10% (*v*/*v*), 2 mL). The organic layer was separated, and the aqueous layer was further extracted with ethyl acetate in hexane (10% (*v*/*v*), 2 mL × 2) using a separatory funnel. The organic layers were combined, washed with deionized water (5 mL), dried over anhydrous sodium sulfate, and concentrated under reduced pressure. Where necessary, the concentrate was diluted with dichloromethane to give a sample volume of 200 µL that was subjected to gas chromatography-mass spectrometry (GC-MS) analysis.

### 4.3. Gas Chromatography-Mass Spectrometry (GC-MS) Analysis

GC-MS analysis was performed on a Shimadzu GCMS QP2010 or Shimadzu GCMS TQ8030 spectrometer equipped with a split/splitless injector, fused silica capillary column (SH-Rtx-5MS, 30 m × 0.25 mm I.D. × 0.25 μm film thickness) with crossbond 5% diphenyl/95% dimethyl polysiloxane as the stationary phase and integrated mass spectrometry (MS). Helium gas (BOC, North Ryde, NSW, Australia) (99.999%) was used as a carrier gas with a constant flow of 1.0 mL/min. The injection was 1 μL of a sample at 270 °C. The initial column temperature was set at 60 °C and held for 4 min, increased to 200 °C at a rate of 10 °C/min, then increased to 300 °C at a rate of 40 °C/min, and held for 7 min. The temperatures of the interface and ion source box were set at 250 and 230 °C, respectively. The ionization method was electron impact at a voltage of 70 eV. The spectra were obtained over a mass range of *m*/*z* 41–650. The data were processed by the Shimadzu GCMS Postrun software. For identification, mass spectra were compared with the NIST library (NIST17-1, NIST17-2, NIST17s) to identify related molecules. The fragmentation patterns and retention indices published in the literature were used to suggest candidate molecules. The proposed molecules were purchased, and GC-MS analyzed using the same instrumental method. The identity of a compound was confirmed by comparing retention time and fragmentations of the authentic molecule. The used solvents, including absolute ethanol, hexane, ethyl acetate, and dichloromethane (DCM) were routinely subjected to GC-MS runs to effectively identify and eliminate impurities.

### 4.4. Quantification of Known Bactrocera Attractants in P. maliformis

The amounts of raspberry ketone, cuelure, zingerone, and zingerol in the collected flower parts were estimated by reference to standard curves. Stock solutions of raspberry ketone, cuelure, zingerone, and zingerol were prepared in volumetric flasks (5 mL), and a series of standard solutions were prepared by serial dilution of the stock solutions. An aliquot of 1.0 μL internal standard solution (tridecane, 1.06 mg/mL) was added to the standards and samples to give a final concentration of 5.30 µg/mL. The standard and sample solutions were run through GC-MS, and the peak area ratios of the analytes to the internal standard were plotted against ratios of known concentrations of the analytes to the internal standard to obtain standard curves. The concentration of a sample was estimated using the slope of the standard curve, and the amounts of these compounds were subsequently estimated using the original volume of a sample for each flower part.

### 4.5. Chromatographic Purification of Unknown Compounds

Following the analyses of the flower parts, all of the corona sample solutions were pooled, and the solvent was evaporated under reduced pressure. The residue was separated by double flash column chromatography with ethyl acetate/hexane (0–20% ethyl acetate in hexane (*v*/*v*), gradient) as a solvent system to give 3-ethyl-2,3-dihydro-4*H*-pyran-4-one and 1-(4-methyl-1,3-dioxolan-2-yl)pentan-3-one, where the chromatographic fractions were monitored by GC. The NMR spectra of the compounds were obtained to support the identities of the compounds.

### 4.6. Data Analysis

The data were not normally distributed, and hence were log transformed for statistical analysis. However, the raw data were used to generate graphs. The change in the amounts of the compounds present in the flower tissues through the day were analyzed by a linear model to see whether the amount of a compound changes with flower collection time. In the analysis, collection time was the independent variable, the detected mass of compound was the dependent variable. Some of the amounts of the compounds were significantly different throughout the day. To compare the amount of a compound between collection times, normally, a posthoc analysis can be followed. Instead, we were interested in decay patterns, and hence further analyzed by a non-linear curve fitting method with the Function (1).
(1)$$f\left(y\right)=A\times e^{-k\times x}+\varepsilon ,$$
where, A = Initial value for x, k = slope, and ε = correction factor. Data were analyzed using R (v3.6.1., R Core Team, 2019).

## Figures and Tables

**Figure 1 molecules-25-02877-f001:**
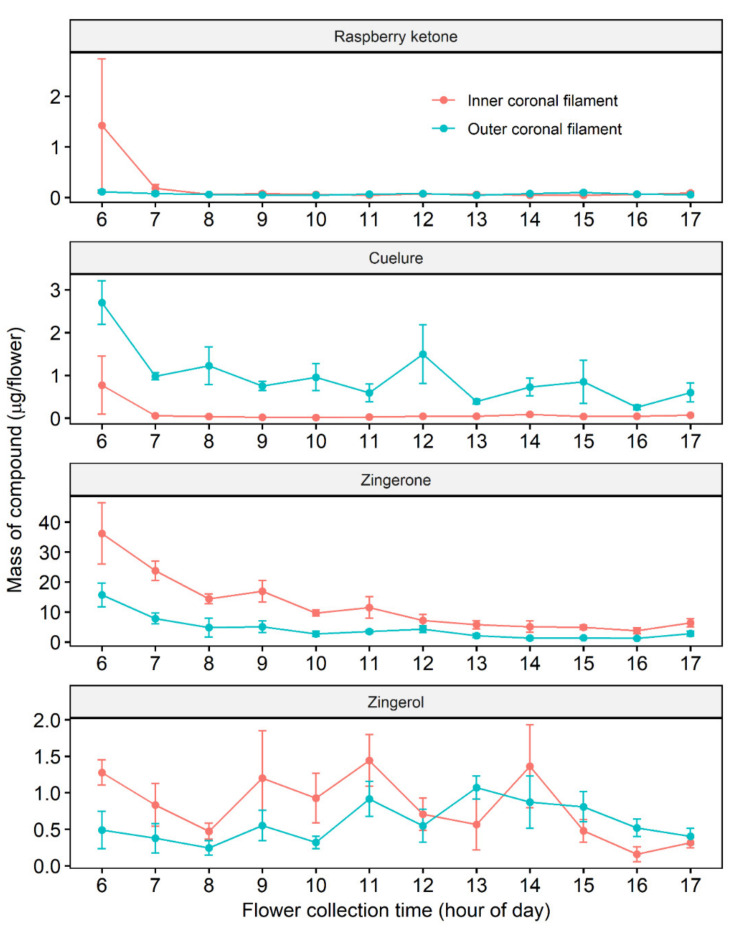
Diurnal changes of raspberry ketone, cuelure, zingerone, and zingerol in the coronal filaments of *P. maliformis*. Error bars represent standard errors that reflect variation across individual flowers.

**Table 1 molecules-25-02877-t001:** Chemical profiles of the seven assessed flower parts of *P. maliformis*. Relative abundances of chemicals were calculated by dividing the gas chromatography (GC) peak area of a compound by the sum of total peak areas of all compounds; The relative abundances were obtained from the 7 am samples; * Tentatively identified compounds, further information is in the Supplementary Material; RI: Kovats retention index; Lit. RI: Literature retention index; ref: references for retention index with most similar GC conditions.; MM: molecular mass; ICF: inner coronal filament; OCF: outer coronal filament; SE: Standard error.

No	Identity	RI	Lit. RI [ref]	MM	Diagnostic Ions	ICF/% (SE)	OCF/% (SE)	Anther/% (SE)	Stigma/% (SE)	Ovary/% (SE)	Petal/% (SE)	Sepal/% (SE)
1	β-Pinene	980	978 [32]	136.2	136, 121, 93, 79, 69, 41	0.0	0.0	0.0	0.0	0.0	26.27 (8.86)	25.39 (9.21)
2	3-Ethyl-2,3-dihydro-4*H*-pyran-4-one *	1042		126.1	126, 98, 81, 70, 53,	5.93 (1.96)	3.43 (1.08)	0.0	1.19 (0.74)	1.21 (0.77)	1.21 (0.69)	1.78 (1.00)
3	Borneol	1175	1171 [33]	154.2	154, 139, 123, 110, 95, 55, 41	0.0	0.0	0.0	0.52 (0.12)	0.38 (0.06)	0.49 (0.11)	0.50 (0.20)
4	1-(4-Methyl-1,3-dioxolan-2-yl)pentan-3-one *	1227		172.2	172, 143, 128, 99, 85, 82, 72, 57	8.91 (2.71)	5.77 (2.21)	2.57 (1.66)	8.38 (3.33)	8.28 (3.43)	4.78 (2.00)	3.22 (1.74)
6	Nonanoic acid	1278	1275 [34]	158.2	129, 115, 73, 60, 57, 41	0.20 (0.04)	0.12 (0.05)	1.03 (0.82)	0.52 (0.14)	0.53 (0.14))	0.42 (0.07)	0.10 (0.04)
7	Indole	1295	1294 [35]	117.1	117, 90	0.16 (0.01)	0.13 (0.04)	0.09 (0.05)	0.0	0.09 (0.06)	0.19 (0.05)	0.18 (0.06)
8	4-(2-Hydroxyisopropyl) -1-methylcyclohexanol (Terpin)	1315		172.3	139, 96, 81, 59, 43	0.0	0.0	2.02 (1.04)	1.22 (0.63)	0.0	0.0	1.04 (0.71)
9	4-(4-Hydroxyphenyl)-2-butanone (Raspberry ketone)	1560	1556 [36]	164.2	164, 107	0.22 (0.04)	0.11 (0.09)	0.0	0.0	0.0	0.05 (0.04)	0.06 (0.03)
10	4-(4-Acetoxyphenyl)-2-butanone (Cuelure)	1652		206.2	206, 164, 107	0.16 (0.06)	1.11 (0.30)	0.0	0.0	0.0	0.03 (0.02)	0.04 (0.02)
11	4-(4-Hydroxy-3-methoxyphenyl)-2-butanone (Zingerone)	1663		194.2	194, 137	1.10 (0.24)	0.12 (0.07)	0.0	0.0	0.0	0.06 (0.02)	0.06 (0.02)
12	4-(4-Hydroxy-3-methoxyphenyl)-2-butanol (Zingerol)	1688		196.2	196, 137	0.22 (0.04)	0.0	0.0	0.0	0.0	0.0	0.0
13	Propyl laurate	1690	1685 [37]	242.4	242, 201, 183, 115, 102, 73, 61, 43	0.0	0.0	0.0	1.88 (1.00)	0.0	0.0	0.0
14	(2*E*,6*E*)-3,7,11-Trimethyldodeca-2,6,10-trien-1-ol (Farnesol)	1732	1728 [38]	222.4	222, 136, 121, 107, 93, 81, 69, 41	5.87 (3.22)	3.92 (1.45)	0.0	0.0	0.0	0.0	0.0
15	3,7,11-Trimethyl-2,6,10-dodecatrien-1-ol acetate (Farnesyl acetate)	1849	1846 [39]	264.4	264, 204, 161, 136, 121, 107, 93, 81, 69, 43	7.78 (2.34)	8.31 (2.45)	0.42 (0.18)	1.76 (1.10)	1.02 (0.47)	0.58 (0.20)	0.52 (0.22)
16	Palmitic acid	1978	1977 [40]	256.4	256, 312, 185, 171, 157, 129, 73, 57	8.76 (1.07)	17.82 (5.78)	12.27 (4.84)	12.52 (5.02)	13.47 (5.62)	7.78 (3.12)	8.42 (3.88)
17	Ethyl palmitate	1997	1996 [41]	284.5	284, 241, 157, 101, 88, 70, 55	6.87 (1.17)	2.33 (0.43)	19.08 (6.66)	14.89 (5.98)	15.08 (6.13)	4.85 (1.96)	5.32 (2.24)
18	Phytol	2121	2117 [41]	296.5	123, 111, 97, 81, 71m, 69, 57	0.23 (0.06)	0.24 (0.09)	0.08 (0.03)	3.86 (2.02)	5.03 (2.38)	1.82 (0.98)	11.45 (4.44)
19	Linoleic acid	2157	2156 [42]	280.4	280, 150, 138, 124, 109, 95, 81, 67, 55	2.38 (0.62)	2.10 (0.64)	8.28 (3.35)	4.42 (2.56)	5.86 (2.18)	4.33 (1.94)	5.78 (2.21)
20	Linolenic acid	2162	2162 [34]	278.4	278, 222, 149, 135, 121, 108, 93, 95, 79, 67, 55	1.83 (0.76)	4.32 (1.76)	2.22 (1.08)	5.11 (2.88)	4.63 (2.23)	2.82 (1.21)	3.61 (1.22)
21	Ethyl linoleate	2171	2171 [43]	308.5	308, 262, 109, 95, 81, 67, 55	9.54 (2.24)	15.23 (4.63)	18.41 (6.33)	19.45 (7.21)	17.94 (6.12)	16.36 (5.05)	16.50 (5.55)
22	Ethyl linolenate	2174	2173 [44]	306.5	306, 264, 149, 135, 121, 108, 95, 79, 67, 55	2.03 (1.34)	2.21 (1.46)	23.77 (7.45)	9.52 (3.32)	9.44 (3.33)	3.70 (1.83)	3.82 (2.00)
23	Stearic acid	2188	2188 [45]	284.5	284, 241, 185, 129, 73, 60, 43	1.84 (0.78)	2.10 (1.05)	1.12 (0.32)	3.83 (1.23)	6.52 (2.28)	1.84 (0.92)	1.79 (1.04)
24	Ethyl sterate	2198	2197 [46]	312.5	312, 269, 213, 157, 101, 88, 670, 55	2.13 (0.83)	1.43 (0.93)	4.03 (2.72)	6.02 (3.44)	4.77 (1.75)	0.72 (0.32)	1.02 (0.41)
25	Squalene	2834	2833 [41]	410.7	367, 341, 203, 175, 161, 136, 121, 85, 81, 69	33.83 (10.01)	32.58 (10.21)	4.49 (2.15)	4.74 (2.83)	5.65 (2.71)	19.85 (7.02)	9.35 (4.44)

**Table 2 molecules-25-02877-t002:** Parameters from the curve fitting results for raspberry ketone, zingerone and cuelure. Fitted function: $f\left(y\right)=A\times e^{-k\times x}+\varepsilon$; A = Initial value for x (collection time); SE: standard error; k = slope; ε = correction factor; T_1/2_ = estimated half-life; RSS = residual sum-of-squares; RSE = residual standard error.

Compound	Flower Part	A (SE)	ε (SE)	k (SE)	T_1/2_, Hour	RSS (RSE)
Raspberry ketone	Inner coronal filament	3429000 (6557000)	0.060 (0.116)	2.456 (3.662)	0.282	34.62 (0.793)
Cuelure	Outer coronal filament	4200 (245700)	0.768 (0.112)	1.667 (1.789)	0.964	31.51 (0.756)
Zingerone	Inner coronal filament	365 (259)	4.807 (2.127)	0.412 (1.331)	0.416	3454 (7.925)
	Outer coronal filament	968 (1166)	2.247 (0.575)	0.719 (1.307)	1.682	563.4 (3.200)

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
