# Peer review of "Zingerone in the Flower of Passiflora maliformis Attracts an Australian Fruit Fly, Bactrocera jarvisi (Tryon)"

_molecules, 2020, doi:10.3390/molecules25122877_

Round 1
Reviewer 1 Report
The publication included chemical composition determinartio of the inner and 17 outer coronal filaments, anther, stigma, ovary, sepal, and petal of P. maliformis were undertaken at different time points. This is a considerable amount of work which was undertaken thoroughly. Standards were used to assist assignation of the compounds and this included a synthesis which was very well performed and characterised. The GCMS work looked to be undertaken well.
The paper was written extremely well written and easy to read. In my view it is eminently publishable.
Author Response
We thank the reviewer for the kind comments
Reviewer 2 Report
The manuscript by Park et al. addresses an interesting topic on a unique interaction between a passionflower and an Australian fruit fly species. The report is well written and the results are solid. I found some minor problems in the experimental design, nomenclature and references as given below by line number.
L 31. methyl eugenol numbering seems incorrect: 1-allyl-3,4-dimethoxybenzene (not -2,4-..); preferably 1,2-dimethoxy-4-(2-propenyl)benzene (?)
L 77. Phenylpropanoids instead of phenyl propanoids
L 148. The occurrence of cuelure in an orchid species has been reported recently:
Katte, T., Tan, K.H., Su, Z.H., Ono, H., and Nishida, R. 2019. Floral fragrances in two closely related fruit fly orchids, Bulbophyllum hortorum and B. macranthoides (Orchidaceae): assortments of phenylbutanoids to attract tephritid fruit fly males. Appl. Entomol. Zool. 55: 55–64.
L 182 and Table 1. Farnesyl acetate: a (2E,6E)-isomer?
L 225. "An aliquot of 1.0 µL internal standard solution..." Technically, at least a 10.0 µL internal standard solution may be required to obtain a reasonable quantification values (as in Table 2) to avoid a large standard error, unless the authours used any special microsyringe device?
It may be a good idea to present a photo of the passionflower as a suppl material, showing the position of filament organs - maybe with the Australian flies visiting?

Author Response
We thank the reviewer for the invaluable comments.
L31 methyl eugenol numbering seems incorrect: 1-allyl-3,4-dimethoxybenzene (not -2,4-..); preferably 1,2-dimethoxy-4-(2-propenyl)benzene (?)
We changed 1-allyl-2,4-dimethoxybenzene to 1-allyl-3,4-dimethoxybenzene.
L 77. Phenylpropanoids instead of phenyl propanoids
We removed a space between phenyl and propanoids
L 148. The occurrence of cuelure in an orchid species has been reported recently:
We cited the paper (Katte et al, 2020, doi: https://doi.org/10.1007/s13355-019-00653-x) as Ref No 71 in L148
L 182 and Table 1. Farnesyl acetate: a (2E,6E)-isomer?
We inserted (mixture of isomers) in L 184. The isomers co-elude in GC and hence we retained farnesyl acetate as it is in Table 1.
L 225. "An aliquot of 1.0 µL internal standard solution..." Technically, at least a 10.0 µL internal standard solution may be required to obtain a reasonable quantification values (as in Table 2) to avoid a large standard error, unless the authours used any special microsyringe device?
Although a larger volume of an aliquot will minimize errors, we used 1.0 uL capacity microsyringe that is specially designed for small quantity injections (even less than 1.0 uL) and hence the errors that might have caused by delivering internal standard solution in the present study are in an acceptable range.
It may be a good idea to present a photo of the passionflower as a suppl material, showing the position of filament organs - maybe with the Australian flies visiting?
We added a photo with the flower and B. jarvisi males visiting in Figure S4 in Supplementary Material and also added “(see Figure S4 in Supplementary Material for the organization of the flower organs)” in L195-196.
Reviewer 3 Report
- Why Authors chosen this kind of column: „fused silica capillary column (30 m x 0.25 mm I.D. x 0.25 μm film thickness)“?
- These studies require a correlation of B. jarvisi activity with P. maliformis flowers in the future. Because the statements that "The diurnal patterns of zingerone and cuelure contents of P. maliformis flowers coincide with the typical diurnal pattern of responsiveness of Bactrocera males, which are usually more responsive to phenylpropanoid attractants in the early morning [29-31]" is not detailed enough. The authors should write what other studies need to be done to correlate the relationship between the activity of flies and the diurnal changes of plant fragrance substances.

Author Response
We thank the reviewer for the invaluable comments.
- Why Authors chosen this kind of column: „fused silica capillary column (30 m x 0.25 mm I.D. x 0.25 μm film thickness)“?
We used a fused silica capillary column (SH-Rtx-5MS) because it is inert, low-bleed and ideal for GC-MS analysis of a range of compounds, such as semi-volatiles, aromatic compounds or polyisoprenoids. The polarity of the column was chosen after we examined 5% and 35% diphenyl phases and found more compounds co-eluding when using the 35% column. The chosen column ID and thickness are suitable to accommodate a typical mass of each analyte in a sample and our samples were in such a range. As this column is typical for the work we carried out, we do not feel that the paper requires a section devoted to the justification of the use of this column. However, to add more clarity, we added “SH-Rtx-5MS” and “crossbond” in L208-209.
- These studies require a correlation of jarvisiactivity with P. maliformis flowers in the future. Because the statements that "The diurnal patterns of zingerone and cuelure contents of P. maliformis flowers coincide with the typical diurnal pattern of responsiveness of Bactroceramales, which are usually more responsive to phenylpropanoid attractants in the early morning [29-31]" is not detailed enough. The authors should write what other studies need to be done to correlate the relationship between the activity of flies and the diurnal changes of plant fragrance substances.
We added “Further study is needed to confirm whether peak responsiveness of B. jarvisi to phenylpropanoids corresponds with timing of greatest amount of phenylpropanoids in P. maliformis.” in L165